# Active, anytime-valid risk controlling prediction sets

**Ziyu Xu**[*]
Department of Statistics and Data Science
Carnegie Mellon University
xzy@cmu.edu

**Nikos Karampatziakis**
Microsoft
nikosk@microsoft.com

**Paul Mineiro**
Microsoft
pmineiro@microsoft.com

## Abstract

Rigorously establishing the safety of black-box machine learning models concerning critical risk measures is important for providing guarantees about model behavior. Recently, Bates et. al. (JACM '24) introduced the notion of a *risk controlling prediction set (RCPS)* for producing prediction sets that are statistically guaranteed low risk from machine learning models. Our method extends this notion to the sequential setting, where we provide guarantees even when the data is collected adaptively, and ensures that the risk guarantee is anytime-valid, i.e., simultaneously holds at all time steps. Further, we propose a framework for constructing RCPSes for active labeling, i.e., allowing one to use a labeling policy that chooses whether to query the true label for each received data point and ensures that the expected proportion of data points whose labels are queried are below a predetermined label budget. We also describe how to use predictors (i.e., the machine learning model for which we provide risk control guarantees) to further improve the utility of our RCPSes by estimating the expected risk conditioned on the covariates. We characterize the optimal choices of label policy and predictor under a fixed label budget and show a regret result that relates the estimation error of the optimal labeling policy and predictor to the wealth process that underlies our RCPSes. Lastly, we present practical ways of formulating label policies and empirically show that our label policies use fewer labels to reach higher utility than naive baseline labeling strategies on both simulations and real data.

## 1   Introduction

One of the core problems of modern deep learning systems is the lack of rigorous statistical guarantees one can ensure about the performance of a model in practice. In particular, we are interested in ensuring the safety of a deep learning system so that it does not incur undue risk while optimizing for an objective of interest. This type of guarantee arises in many applications. For example, a deep learning based medical imaging segmentation system that detects lesions [36, 10, 29] should guarantee that it does not miss most of the lesion tissue while remaining precise and minimizing the total amount of tissue that is highlighted. Hence, it is crucial to provide a statistical guarantee about the "safety" of any machine learning system to be deployed. Bates et al. [4] introduced the notion of a *risk controlling prediction set* as a method to derive such guarantees on top of the outputs of a wide range of black-box models. They consider the setting where all the calibration data for verifying statistical safety guarantees is available before deployment and the model only needs to be

---

[*]Part of work done while interning at Microsoft.

38th Conference on Neural Information Processing Systems (NeurIPS 2024).

calibrated once, i.e., the batch setting. However, this is often unrealistic in a production setup when we have no data concerning the performance of a model on the distribution of interest before we deploy it, and we wish to update our calibration each time a new data point (or group of data points) arrives. Consequently, it is natural to calibrate the machine learning model in an *online* fashion while receiving new data sequentially. Unfortunately, methods for obtaining statistical guarantees in the batch setting of Bates et al. [4] do not ensure risk control guarantees in the sequential regime. Further, when one uses data from production, the raw data is unlabeled, and one must expend resources (either paying experts or utilizing a more powerful model) to create gold labels for these data points. Hence, we generalize the sequential setup to an active setting in which we see the covariate $X$ (e.g., an image, a natural language query from the user, etc.) and choose whether to query the true label $Y$. Concretely, consider the following scenarios where an active and sequential method is relevant.

- *Reduce query cost in medical imaging.* A medical imaging system that outputs scores for for each pixel of image that determines whether there is a lesion or not would want to utilize labels given by medical experts for unlabeled images from new patients. Since the cost of asking experts to label these images is quite high, one would want to query experts efficiently, and only on data that would be most helpful for reducing the number of highlighted pixels.

- *Domain adaptation for behavior prediction.* Often, the post-deployment distribution is different from that which has been seen before. For example, during a navigation task for a robot, we may want to predict the actions of other agents and avoid colliding into them when travelling between two points [20]. Since agents may behave differently in every environment, it makes sense to collect the behavior data in the test environment and update the behavior prediction in an online fashion to get accurate predictions calibrated for that specific environment.

- *Safe outputs for large language models (LLMs).* One of the goals with large language models is to ensure their responses are not harmful in some fashion (e.g., factually wrong, toxic, etc.). One can view this as producing a prediction set for the binary label set of $Y \in \{\texttt{harmful}, \texttt{not harmful}\}$. Many pipelines for modern LLMs include some form of a safety classifier, which scores the risk level of an output, and determines whether it should be output to the user or not [21, 15], or a default backup response should be used instead. One would want to label production data acquired from user interaction with the LLM and used to calibrate cutoff for the scores that are considered low enough for the response to be allowed through.

**Example: image classification** Let us assume we wish to classify an image $X \in \mathcal{X}$, and we have access to a probabilistic classifier $s : \mathcal{X} \to \Delta^{\mathcal{Y}}$ where $\Delta^{\mathcal{Y}}$ is the probability simplex over distributions over all possible classes, $\mathcal{Y}$. Let $s^y(x)$ denote the probability of class $y$ in the distribution $s(x)$. Based on the probabilities from $s(X)$, we can define $C(X, \beta)$ to have the labels with the largest probabilities that sum to $\beta \in [0, 1]$ in the following fashion:

$$\gamma(X, \beta) := \max \left\{ \gamma \in [0, 1] : \sum_{y \in \mathcal{Y}} \mathbf{1}\left\{ s^y(X) \geq \gamma \right\} \cdot s^y(X) \geq \beta \right\},$$

$$C(X, \beta) := \{ y \in \mathcal{Y} : s^y(X) \geq \gamma(X, \beta) \}$$

Now, we can define the miscoverage error of our label set $C(X, \beta)$ as follows:

$$r(X, Y, \beta) := \mathbf{1}\left\{ Y \notin C(X, \beta) \right\}. \tag{1}$$

Now, assume that $(X, Y) \sim \mathcal{P}^*$, i.e., the images and class are jointly drawn from a fixed distribution. We want to find a choice of $\beta$ such that $\rho(\beta) := \mathbb{E}[r(X, Y, \beta)]$ is guaranteed to be at most $\theta \in [0, 1]$, i.e., the expected miscoverage over the population of images and labels is at most $\theta$.

In the above image classification example, we do not simply wish to find any $\beta$ that ensures $\rho(\beta) \leq \theta$ — setting $\beta = 1$ would trivially ensure this guarantee for any $\theta \in [0, 1]$. We also want to minimize the size of our uncertainty set $C(X, \beta)$. To present this formulation in more general terms, we are interested in solving the following problem for a fixed level of risk control $\theta \in [0, 1]$:

$$\max_{\beta} g(\beta) \qquad \text{subject to } \rho(\beta) \leq \theta. \tag{2}$$

where $g$ is the utility of our choice. We make the following natural assumption about $r$, $\rho$, and $g$.

*Assumption* 1. $g$ and $\rho$ are monotonically decreasing w.r.t. $\beta$ and we assume $\rho(1) = 0$. In addition, $\rho$ is right-continuous.

Our image classification example has an expected risk and utility that satisfy the respective monotonicity assumptions, and such risk measures arise in many applications such as natural language question answering [26], image segmentation [1], and behavior control for robotics [20, 16]. Assumption 1 implies that maximizing $g(\beta)$ is equivalent to minimizing $\beta$, as $g$ is decreasing in $\beta$, and the right-continuity of $\rho$ allows us to define the notion of an optimal calibration parameter that is the solution to (2):

$$\beta^* := \min \{\beta \in [0, 1] : \rho(\beta) \leq \theta\}.$$

Our goal in this paper is to derive a sequence of upper bounds on $\beta^*$ that quickly approach the true $\beta^*$ but provide "anytime-valid risk control" in the sense that they are always greater the smallest "safe" parameter $\beta^*$ and induce risk under $\theta$, i.e., $\beta \geq \beta^*$ implies that $\rho(\beta) \leq \rho(\beta^*) \leq \theta$. Since we are guaranteed by Assumption 1 that $\rho(1) = 0 \leq \theta$, we always have a safe option of $\beta = 1$ to start with as our upper bound.

**Our contributions.** The primary contributions of this paper are as follows.

1. *Extensions of RCPS to anytime-valid and active settings.* We extend the notion of RCPS in two ways: (1) to enable anytime-valid RCPS which allows one to refine the set as one receives more samples in a stream while maintaining risk control throughout the entire stream, and (2) to define an RCPS that is valid under active learning, i.e., enable us to decide whether to label each example based on the covariates. We also define a way for incorporating risk predictions from the machine learning model to decrease variance further and reduce the number of labels needed to estimate $\beta^*$. We formulate this betting framework in Section 2.

2. *Deriving powerful labeling policies and predictors.* We show in Section 3 that our active, anytime-valid RCPS methods are practically powerful and converge to $\beta^*$ in a label efficient manner by also deriving formulations for the optimal labeling policy and predictors under the standard log-optimality criterion that is used for evaluating anytime-valid methods [13, 34, 19]. We derive explicit regret bounds w.r.t. a lower bound on the growth rate on the wealth processes that underlie our RCPS methods. These bounds characterize how the deviation of any labeling policy and predictor from the log-optimal policy and predictor affect the growth rate of our wealth processes (and hence the earliest time at which a candidate $\beta$ is removed from consideration as $\beta^*$). In Section 4 we also show that machine learning model based estimators of the optimal policy and predictors are label efficient in practice through experiments.

**Related work.** Most relevant to this paper is the recent work from Zrnic and Candès [37] that provides a rigorous framework for statistical inference with active labeling policies, and leverages machine learning predictions through prediction-powered inference [2]. However, their focus is on M-estimation and deriving asymptotic, martingale central-limit theorem based results for a parameter. On the other hand, we provide finite sample anytime-valid results that are also valid at adaptive stopping times that directly utilize e-process [28] construction of sequential tests. Further, our goal is to provide a time-uniform statistical guarantee in the RCPS framework rather than directly estimating a parameter with adaptively collected data. We discuss additional related work in depth in Section 5.

## 2  Anytime-valid risk control through betting

We use $(X_t)_{t \in \mathbb{I}}$ to denote a sequence that is indexed by $t$ with index set $\mathbb{I}$. If the index set or indexing variable is apparent in context, we drop it for brevity. In our setup, we assume that our data points arrive in a stream $(X_1, Y_1), (X_2, Y_2), \ldots$ that proceeds indefinitely. Let $(\mathcal{F}_t)$ be the canonical filtration on the data, i.e., $\mathcal{F}_t := \sigma(\{(X_i, Y_i)\}_{i \leq [t]})$ is the sigma-algebra over the first $t$ points. Recall that we assumed $(X_t, Y_t) \sim \mathcal{P}^*$ are i.i.d. draws for each $t \in \mathbb{N} := \{1, 2, 3, \ldots\}$, and want to control the risk $\rho(\beta) = \mathbb{E}_{(X,Y) \sim \mathcal{P}^*}[r(X, Y, \beta)]$ where the expectation is take only over $(X, Y)$. We illustrate an overview of our methodology (which we describe in the sequel) in Figure 1.

We desire to output a sequence of calibration parameters, $(\widehat{\beta}_t)$, such that every $\beta_t$ is "safe", i.e., ensures that the resulting risk of the output is provably controlled under a fixed level.

*Definition* 1. A sequence of calibration parameters $(\widehat{\beta}_t)$ is said to have $(\theta, \alpha)$-*anytime-valid risk control* if it possesses the following property:

$$\mathbb{P}(\rho(\widehat{\beta}_t) \leq \theta \text{ for all } t \in \mathbb{N}) \geq 1 - \alpha. \tag{3}$$

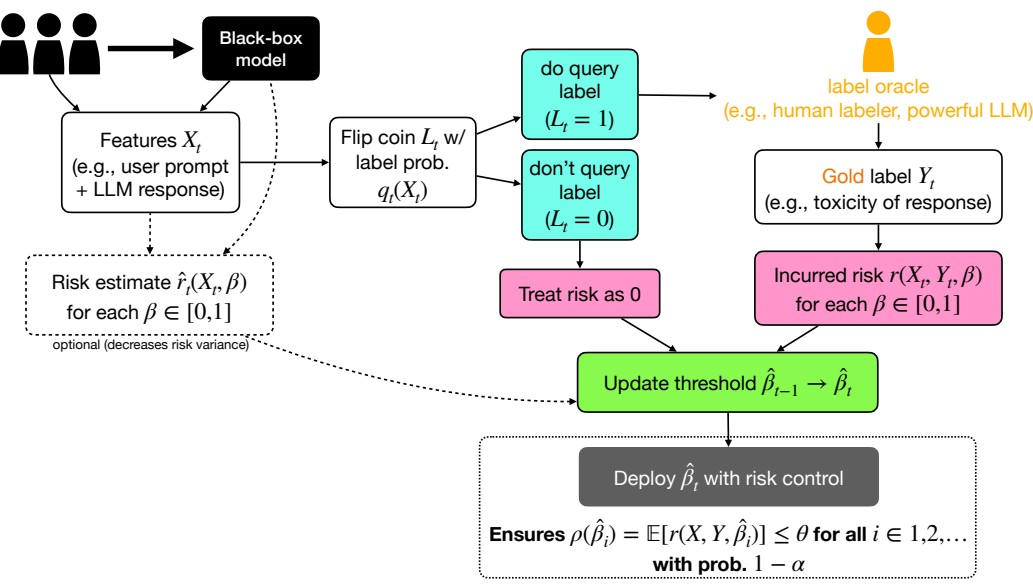

Figure 1: Diagram of the active labeling setup for ensuring anytime-valid risk control.

We name this as "anytime-valid" since the risk control condition (i.e., $\rho(\widehat{\beta}_t) \leq \theta$) is guaranteed to hold simultaneously at all $t \in \mathbb{N}$. Hence, this allows for the user to process a continuous stream of data and control the probability that a $\widehat{\beta}_t$ is chosen at any time $t$ that is "unsafe", i.e., $\rho(\widehat{\beta}_t) > \theta$. We build on recent work that develops a framework for hypothesis testing and parameter estimation with sequential data collection based on martingales and gambling with virtual wealth known as *testing by betting* [30]. In this framework, the goal is to design an *e-process*, $(E_t)$, w.r.t. a null hypothesis $H_0$, which satisfies the following properties when true:

*Definition* 2. An *e-process*, $(E_t)_{t \in \mathbb{N}_0}$, w.r.t. a hypothesis $H_0$, is a nonnegative process for which there exists another nonnegative process, $(M_t)_{t \in \mathbb{N}_0}$ s.t. the following is true when $H_0$ is true: (1) $\mathbb{E}[M_0] \leq 1$, (2) $M_t \geq E_t$ for all $t \in \mathbb{N}$ almost surely and (3) $\mathbb{E}[M_t \mid \mathcal{F}_{t-1}] \leq M_{t-1}$ for all $t \in \mathbb{N}$,, i.e., $(M_t)$ is a supermartingale.

E-processes will be the main tool we use to construct $(\widehat{\beta}_t)$. We leverage the probabilistic bound on e-processes provided by Ville's inequality to prove our anytime-valid risk control guarantee.

*Fact* 1 (Ville's inequality [33]). For any e-process, $(E_t)$, with initial expectation bounded by 1, i.e., $\mathbb{E}[E_0] \leq 1$, we have that

$$\mathbb{P}\left(\text{exists } t \in \mathbb{N} : M_t \geq \alpha^{-1}\right) \leq \alpha \text{ for each } \alpha \in [0, 1].$$

In this paper, all our e-processes will also be nonnegative supermartingales, so we denote them as $(M_t)$. Now, we will specify the null hypotheses in our risk control setting. For each $\beta \in [0, 1]$, we test the null hypothesis $H_0^\beta : \rho(\beta) \geq \theta$ for a fixed risk control level $\theta \in [0, 1]$. Note that we include equality to $\theta$ in the null hypothesis since we do not wish to reject $H_0^{\beta^*}$. Let $\{(M_t(\beta))\}_{\beta \in [0,1]}$ be a family of e-processes where $(M_t(\beta))$ is an e-process for $H_0^\beta$. Then, we can derive $\widehat{\beta}_t$ for each $t \in \mathbb{N}$ as follows:

$$\widehat{\beta}_t := \min \{\beta \in [0, 1] : M_t(\beta') \geq 1/\alpha \text{ for all } \beta' > \beta\}. \tag{4}$$

**Theorem 1.** *The sequence of estimates* $(\widehat{\beta}_t)$ *in* (4) *satisfies the anytime-valid risk control guarantee* (3), *i.e.,* $\mathbb{P}(\rho(\widehat{\beta}_t) \leq \theta \text{ for all } t \in \mathbb{N}) \geq 1 - \alpha$.

*Proof.* First, we note that

$$\{\exists t \in \mathbb{N} : \rho(\widehat{\beta}_t) > \theta\} \Leftrightarrow \{\exists t \in \mathbb{N} : \widehat{\beta}_t < \beta^*\} \Rightarrow \{\exists t \in \mathbb{N} : M_t(\beta^*) \geq 1/\alpha\}.$$

Since $H_0^{\beta^*}$ is always true by definition of $\beta^*$, we get that $(M_t(\beta^*))$ is an e-process by Proposition 1. Thus, by applying Ville's inequality, we get that:

$$\mathbb{P}(\exists t \in \mathbb{N} : \rho(\widehat{\beta}_t) > \theta) \leq \mathbb{P}\left(\exists t \in \mathbb{N} : M_t(\beta^*) \geq 1/\alpha\right) \leq \alpha.$$

$\square$

Now, we will present a concrete example of an e-process. Denote $R_t(\beta) := r(X_t, Y_t, \beta)$. We test $H_0^{\beta}$ using the betting e-process from Waudby-Smith and Ramdas [34]:

$$M_t(\beta) = \prod_{i=1}^{t} \left(1 + \lambda_i \left(\theta - R_i(\beta)\right)\right), \tag{5}$$

where $(\lambda_t)$ is predictable w.r.t. $(\mathcal{F}_t)$, i.e., $\lambda_t$ can be determined by $\mathcal{F}_{t-1}$ for each $t \in \mathbb{N}$, and $\lambda_t \in [0, (1-\theta)^{-1}]$.

**Proposition 1.** $(M_t(\beta))$ in (5) is an e-process for all $\beta$ where $H_0^{\beta}$ is true, i.e., where $\rho(\beta) \geq \theta$.

*Proof.* We note that $(M_t(\beta))$ is nonnegative by the support of $\lambda_t$ being limited, i.e.,

$$1 + \lambda_t(\theta - r(X_t, Y_t, \beta)) \geq 1 + \lambda_t(\theta - 1) \geq 0.$$

Now, we will also show that $(M_t(\beta))$ is a supermartingale when $H_0^{\beta}$ is true.

$$\begin{aligned}
\mathbb{E}[M_t(\beta) \mid \mathcal{F}_{t-1}] &= \mathbb{E}[1 + \lambda_t(\theta - R_t(\beta)) \mid \mathcal{F}_{t-1}] \cdot M_{t-1}(\beta) \\
&= (1 + \lambda_t(\theta - \mathbb{E}[R_t(\beta) \mid \mathcal{F}_{t-1}])) \cdot M_{t-1}(\beta) \leq M_{t-1}(\beta).
\end{aligned}$$

The second equality is because $\lambda_t$ is measurable w.r.t. $\mathcal{F}_{t-1}$ and the last inequality is by $R_t(\beta)$ being independent of $\mathcal{F}_{t-1}$ and $\mathbb{E}[R_t(\beta)] \leq \theta$ being true under $H_0^{\beta}$. Thus, we have our desired result. $\square$

Now, we have a concrete way to derive $(\widehat{\beta}_t)$ that ensures the risk $\rho(\widehat{\beta}_t)$, is controlled at every time step $t \in \mathbb{N}$. However, this requires one to label every example that arrives, i.e., it requires access to entire stream of labels $(Y_t)$. We will now derive a more label efficient way for constructing $(\widehat{\beta}_t)$.

*Remark* 1. Ramdas et al. [27] show that e-processes of the form in (5) characterize the set of admissible e-processes (and hence anytime-valid sequential tests) for testing the mean of bounded random variables. Hence, it is an optimal choice of e-process for our setting, and have been shown to perform better both theoretically and empirically than other sequential tests for bounded random variables (e.g., Hoeffding and empirical-Bernstein based tests [34]).

## 2.1 Active sampling for risk control

Now, we describe active learning for risk control, where an algorithm sees $X_t$ decides whether a label, for the current point, $Y_t$, should be queried or not. At each step $t \in \mathbb{N}$, the algorithm produces a label policy $q_t : \mathcal{X} \to [q_t^{\min}, 1]$ based on the observed data (i.e., $\mathcal{F}_{t-1}$). It then queries the label, $Y_t$, with probability $q_t(X_t)$ that is lower bounded by a constant, $q_t^{\min}$. Let $L_t$ be the indicator random variable for whether the $t$th label is queried, i.e., $L_t \sim \text{Bern}(q_t(X_t))$.

To produce a label efficient method, one would hope to label the most "impactful" data points that result in the largest growth of $M_t(\beta)$ for choices of $\beta \in (0, \widehat{\beta}_t)$, i.e. that are still in consideration for the next $\widehat{\beta}_{t+1}$. For the labeling policies we consider in this paper, we let $q_t^{\min} \in [0, 1]$ be a lower bound on the labeling probability, i.e., $q_t(X_t) \geq q_t^{\min}$ almost surely. Thus, we can derive the following e-process for any sequence of labeling policies $(q_t)$.

$$M_t(\beta) := \prod_{i=1}^{t} \left(1 + \lambda_i \left(\theta - \frac{L_i}{q_i(X_i)} \cdot R_i(\beta)\right)\right). \tag{6}$$

**Proposition 2.** *Let $(q_t)$ be a sequence of labeling policies, and $(\lambda_t)$ be a sequence of betting parameters, and let both sequences be predictable w.r.t. $(\mathcal{F}_t)$ (i.e., $q_t$ and $\lambda_t$ are measurable w.r.t. $\mathcal{F}_{t-1}$ for each $t \in \mathbb{N}$). Then, $(M_t(\beta))$ in (6) is an e-process for all $\beta$ where $H_0^{\beta}$ is true.*

*Proof.* The proof of this is similar to that of inverse propensity-weighted e-processes derived in Waudby-Smith et al. [35]. We know that $(M_t(\beta))$ is nonnegative by the support of $\lambda_t$ being limited:

$$1 + \lambda_t \left( \theta - \frac{L_t}{q_t(X_t)} \cdot R_t(\beta) \right) \geq 1 + \lambda_t \left( \theta - (q_t^{\min})^{-1} \right) \geq 0,$$

where the first inequality is by $R_t(\beta) \leq 1$ and $q_t(X_t) \geq q_t^{\min}$, and the second inequality is by $\lambda_t \leq ((q_t^{\min})^{-1} - \theta)^{-1}$. Now, we will also show that $(M_t(\beta))$ is a supermartingale. We first show the following upper bound:

$$\mathbb{E}\left[ \frac{L_t}{q_t(X_t)} \cdot R_t(\beta) \mid \mathcal{F}_{t-1} \right] = \mathbb{E}\left[ \frac{\mathbb{E}[L_t \mid X_t, \mathcal{F}_{t-1}]}{q_t(X_t)} R_t(\beta) \mid \mathcal{F}_{t-1} \right] = \mathbb{E}[R_t(\beta) \mid \mathcal{F}_{t-1}] \leq \theta. \quad (7)$$

The first equality is by further conditioning on $X_t$, and the second equality is since $L_t$ is defined to be a Bernoulli random variable with parameter $q_t(X_t)$ that is independent of all other randomness when conditioned on $X_t$ and $\mathcal{F}_{t-1}$. The last inequality is by $H_0^\beta$ being true. Now, we have that

$$\mathbb{E}[M_t(\beta) \mid \mathcal{F}_{t-1}] = \left( 1 + \lambda_t \left( \theta - \mathbb{E}\left[ \frac{L_t}{q_t(X_t)} R_t(\beta) \mid \mathcal{F}_{t-1} \right] \right) \right) M_{t-1}(\beta) \leq M_{t-1}(\beta),$$

where the inequality is by (7). Hence, we have shown that $(M_t(\beta))$ is a nonnegative supermartingale, and hence also an e-process, under $H_0^\beta$. $\qquad\square$

**Theorem 2.** $(\widehat{\beta}_t)$ *defined w.r.t.* (6) *satisfies the anytime-valid risk control guarantee* (3).

This is a result of Proposition 2 and Ville's inequality, similar to the proof of Theorem 1. Theorem 2 essentially shows that we can still design e-processes by allowing for a probabilistic label policy.

## 2.2 Variance reduction through prediction

Often, we also have an estimate of the risk we incur, e.g., in the example given for classification, we have an estimated probability distribution over possible outcomes. As a result, we also have an empirical estimate of $\mathbb{E}[r(X, Y, \beta) \mid X = x]$ for each $\beta \in [0, 1]$ that we can use to reduce the variance of our estimate. This is similar to the usage of control variates for improving Monte Carlo estimation [3, § V.2], and of predictors in the recently formulated prediction-powered inference framework [2]. Let $\widehat{r}_t : \mathcal{X} \times [0, 1] \to [0, 1]$ be an estimator of the risk incurred by parameter $\beta$ conditional on $x \in \mathcal{X}$ for each time step $t \in \mathbb{N}$. $(\widehat{r}_t)$ is predictable w.r.t. $(\mathcal{F}_t)$.

**Where does $\widehat{r}$ come from?**   We note that often machine learning models have some estimate $\hat{P}(X)$ of the conditional distribution of $Y \mid X$ (e.g, class probabilities, conditional diffusion models, LLMs, etc.). Thus, for any realized covariate $x$, we can derive use $\mathbb{E}_{Y \sim \hat{P}(x)}[r(X, Y, \beta) \mid X = x]$ from the machine learning model as our choice of $\widehat{r}(x, \beta)$. This expectation can either be calculated analytically (as we do in or classification examples in our experiments) or derived using Monte Carlo approximation (for generative models such as LLMs, one can sample from the conditional distribution). In essence, we can obtain a predictor from the very model we are calibrating. We may also update our predictor using new $(X_t, Y_t)$ pairs we receive for calibrating $(\widehat{\beta}_t)$.

Now, we define our e-process that utilizes our predictor as follows:

$$M_t(\beta) \coloneqq \prod_{i=1}^{t} \left( 1 + \lambda_i \left( \theta - \widehat{r}_i(X_i, \beta) - \frac{L_i}{q_i(X_i)} \cdot \bar{R}_i(\beta) \right) \right) \text{ where } \bar{R}_t(\beta) \coloneqq R_t(\beta) - \widehat{r}_t(X_t, \beta),$$
$$(8)$$

and we restrict $\lambda_t \in [0, ((q_t^{\min})^{-1} - \theta)^{-1}]$ (or in other words, $q_t^{\min} \geq \lambda_t/(1 + \lambda_t\theta)$). Note that this e-process recovers the active e-process defined in (6) if we set $\widehat{r}_t(\cdot, \beta) = 0$ for all $t \in \mathbb{N}$.

**Proposition 3.** $(M_t(\beta))$ *as defined in* (8) *is an e-process for* $H_0^\beta$.

*Proof.* Since the restriction on $(\lambda_t)$ ensures $M_t(\beta)$ is nonnegative, to show that $(M_t(\beta))$ is an e-process, it is sufficient to show:

$$\mathbb{E}[\lambda_t(\theta - \widehat{r}_t(X_t, \beta) - L_t \cdot q_t(X_t)^{-1} \cdot \bar{R}_t(\beta)+) \mid \mathcal{F}_{t-1}]$$
$$= \lambda_t(\theta - \mathbb{E}[\widehat{r}_t(X_t, \beta) + L_t \cdot q_t(X_t)^{-1} \cdot \bar{R}_t(\beta) \mid \mathcal{F}_{t-1}])$$
$$= \lambda_t(\theta - \mathbb{E}[\widehat{r}_t(X_t, \beta) + \mathbb{E}[L_t \cdot q_t(X_t)^{-1} \mid X_t, \mathcal{F}_{t-1}] \cdot \bar{R}_t(\beta) \mid \mathcal{F}_{t-1}])$$
$$= \lambda_t(\theta - \mathbb{E}[\widehat{r}_t(X_t, \beta) + \bar{R}_t(\beta) \mid \mathcal{F}_{t-1}]) = \lambda_t(\theta - \mathbb{E}[R_t(\beta) \mid \mathcal{F}_{t-1}]) \leq 0.$$

The 3rd equality is by definition of $L_t$, the last equality by the definition of $\bar{R}_t$, and the last inequality is due to $H_0^\beta$ being true. $\qquad\square$

The role of $\widehat{r}_t(X_t, \beta)$ is to accurately predict $R_t(\beta)$. Bad predictions can increase the variance of $\bar{R}_r(\beta)$ and lead to slower growth of $M_t(\beta)$, but do not compromise the risk control guarantee. On the other hand, accurate predictions, which come from pretrained models, decrease variance and improve the growth of $M_t(\beta)$. We characterize the optimal predictor (Proposition 4) and relate the accuracy of a predictor to its effect on the e-process (Theorem 3) in the next section.

## 3 Optimal labeling policies

Since the goal of having an active labeling policy is to label fewer data points, one reasonable way of doing this is to maximize the growth rate of our e-process $(M_t(\beta))$ defined in (8). Define the following function, for some $\beta \in [0, 1]$, of a labeling policy $q$, predictor $\widehat{r}$, and betting parameter $\lambda$ where we let $L \mid X \sim \mathrm{Bern}(q(X))$ and $(X, Y) \sim \mathcal{P}^*$:

$$G^\beta(q, \widehat{r}, \lambda) := \log\left(1 + \lambda\left(\theta - \widehat{r}(X, \beta) - \frac{L}{q(X)}\bar{R}(\beta)\right)\right) \text{ where } \bar{R}(\beta) := r(X, Y, \beta) - \widehat{r}(X, \beta).$$

Define the *growth rate* at the $t$th step of $(M_t(\beta))$ as $\mathbb{G}_t^\beta := \mathbb{E}[G_t^\beta(q_t, \widehat{r}_t, \lambda_t)]$, where we let $G_t^\beta$ be identical to $G_t$ but with $X$ and $Y$ replaced with $X_t$ and $Y_t$, respectively. It is a standard notion of power or sample efficiency for e-processes. Typically, our goal when designing an e-process based test is to maximize such a metric, i.e., we want our e-process to be log-optimal [13, 34, 19]. Log-optimality is also called the Kelly criterion in finance [18] and it is known that maximizing the growth rate of a process is equivalent to minimizing the expected time for the process to exceed a threshold, i.e., for our sequential test to reject a value of $\beta$, in the limit as the threshold approaches infinity [5]. Thus, in an asymptotic sense, maximizing the growth rate is equivalent to minimizing the expected time for rejection. Our goal is to maximize the growth rate while having a constraint on the number of labels we can produce.

Let $B \in [0, 1]$ be the constraint on our labeling budget, i.e., we label, in expectation, a $B$ fraction of all data points that we receive. To achieve both of these goals, we wish to choose $q_t, \widehat{r}_t$, and $\lambda_t$ that are the solutions to the following optimization problem:

$$\max_{q, \widehat{r}, \lambda} \mathbb{E}_{L \sim q(X)}[G^\beta(q, \widehat{r}, \lambda)] \text{ s.t. } \mathbb{E}[q(X)] \le B.$$

Since solving the above optimization problem is analytically difficult, one can instead maximize a lower bound on the expected growth [32, 25, 24]:

$$\widehat{G}^\beta(q, \widehat{r}, \lambda) := \lambda\left(\theta - \widehat{r}(X, \beta) - \frac{L}{q(X)}\bar{R}(\beta)\right) - \lambda^2\left(\theta - \widehat{r}(X, \beta) - \frac{L}{q(X)}\bar{R}(\beta)\right)^2$$

$$\le G^\beta(q, \widehat{r}, \lambda), \tag{9}$$

which holds when $\lambda \in [0, (2(q^{\min})^{-1} - 2\theta)^{-1}]$, where $q^{\min} := \inf_{x \in \mathcal{X}} q(x)$. We can further simplify We can use the lower bound in (9) to formulate the following optimization problem.

$$\max_{q, \widehat{r}, \lambda} \mathbb{E}\left[\widehat{G}^\beta(q, \widehat{r}, \lambda)\right] \text{ s.t. } \mathbb{E}[q(X)] \le B \tag{10}$$

Let $(q^*, r^*, \lambda^*)$ be the tuple that is the solution to (10). We can analytically show what $r^*$ is.

**Proposition 4.** *The optimal predictor $r^*$ in the solution to (10) is $r^*(x, \beta) = \mathbb{E}[r(X, Y, \beta) \mid X = x]$ for each $x \in \mathcal{X}$.*

We defer the proof to Appendix A.1. The optimal choice of $q^*$ has the following formulation.

**Proposition 5.** *If we fix $\widehat{r}$ and $\lambda$, the solution to the optimization problem in (10) is given by $q_\beta^*$ where $q_\beta^*(x) \propto \sqrt{\mathbb{E}[\bar{R}(\beta)^2 \mid X = x]}$ for each $x \in \mathcal{X}$ if such a $q_\beta^*$ exists.*

We defer the proof to Appendix A.2. Let $\sigma_\beta(x) := \sqrt{\mathbb{V}[r(X, Y, \beta) \mid X = x]}$ be the conditional standard deviation of $r(X, Y, \beta)$. Now, we can argue that the solution to the optimization problem on the growth rate lower bound in (10) has the following characterization.

*Corollary* 1. The optimal choice of $q_\beta^*$ and $\lambda^*$ that solves (10) is

$$q_\beta^*(x) := \frac{\sigma_\beta(x)}{\mathbb{E}[\sigma_\beta(X)]} \cdot B, \qquad \lambda^* := \frac{1}{2} \cdot \frac{\theta - \rho(\beta)}{(\theta - \rho(\beta))^2 + \mathbb{E}[\sigma_\beta(X)]^2 \cdot B^{-1} + \mathbb{V}[r(X, Y, \beta)]}$$

if $q_\beta^*(x) \in [0, 1]$ for all $x \in \mathcal{X}$, and $\lambda^* \leq (2(\inf_{x \in \mathcal{X}} q_\beta^*(x))^{-1} - 2\theta)^{-1}$. The resulting growth rate has the following lower bound:

$$\mathbb{E}[G_t^\beta(q^*, r^*, \lambda^*)] \geq \mathbb{E}[\widehat{G}_t^\beta(q^*, r^*, \lambda^*)] = \frac{1}{4} \cdot \frac{(\theta - \rho(\beta))^2}{(\theta - \rho(\beta))^2 + \mathbb{E}[\sigma_\beta(X)]^2 \cdot B^{-1} + \mathbb{V}[r(X, Y, \beta)]}$$

We can show this is true as a consequence of Proposition 4, Proposition 5, and solving the quadratic equation that arises for the growth rate to derive the optimal choice of $\lambda^*$. Further, we note that we can define regret of a sequence $(\lambda_t)$ compared to $\lambda^*$ on $\widehat{G}_t^\beta$ as follows.

*Definition* 3. The $\widehat{G}^\beta$-*regret* at the $t$th step of a sequence of betting parameters $(\lambda_t)$ for a risk upper bound $\theta \in [0, 1]$, and a sequence of labeling policies $(q_t)$ and predictors $(\widehat{r}_t)$ where $q_t(x) \geq \varepsilon > 0$ for all $x \in \mathcal{X}$ and $t \in \mathbb{N}$ almost surely is defined as follows:

$$\text{Reg}_t := \max_{\lambda \in [0, (2\varepsilon^{-1} - 2\theta)^{-1}]} \sum_{i=1}^t \mathbb{E}[\widehat{G}_t^\beta(q_t, \widehat{r}_t, \lambda) \mid \mathcal{F}_{t-1}] - \mathbb{E}[\widehat{G}_t^\beta(q_t, \widehat{r}_t, \lambda_t) \mid \mathcal{F}_{t-1}].$$

Since $\widehat{G}_t^\beta(q, \widehat{r}, \lambda)$ is exp-concave in $\lambda$, existing online learning algorithms such as Online Newton Step (ONS) [8] can get $o(T)$ regret guarantees, which means that the growth rate of $(\lambda_t)$ averaged over time will approach (or exceed) the optimal growth rate under $\lambda^*$. For simplicity of analysis, we make the following assumption about the labeling probability of the optimal policy, $q_\beta^*$.

*Assumption* 2. Let $\varepsilon > 0$ be a positive constant. Assume that $q_\beta^*(x) \geq \varepsilon$ for each $x \in \mathcal{X}$.

The lower bound in the above assumption is an analog of the propensity score lower bound on optimal policies that is needed for proving valid inference in adaptive experimentation [17, 7]. Further, we do not need this assumption to hold on every $\beta$, since we are not necessarily interested in log-optimality w.r.t. fringe $\beta$ that are quite far away from $\beta^*$ — in practice having this assumption hold for values of $\beta$ near $\beta^*$ suffices to develop an estimator $\widehat{\beta}$ that shrinks toward $\beta^*$ quickly. Now, we describe how much the growth rates deviates based on on how well $q^*$ and $r^*$ are estimated.

**Theorem 3.** *Let $(\lambda_t)$ be a sequence with $\widehat{G}^\beta$-regret $(\text{Reg}_t)$ and $(q_t)$ and $(\widehat{r}_t)$ are sequences of labeling policies and predictors that are all predictable w.r.t. $(\mathcal{F}_t)$. For a positive constant $\varepsilon > 0$, let $q_t(x) \geq \varepsilon > 0$ for each $t \in \mathbb{N}$ and $x \in \mathcal{X}$ almost surely. Under Assumption 2 for the same $\varepsilon$, the following bound holds:*

$$\sum_{i=1}^t \mathbb{E}[\widehat{G}_t^\beta(q^*, r^*, \lambda^*) - \widehat{G}_t^\beta(q_t, \widehat{r}_t, \lambda_t)] \leq \text{Reg}_t + \sum_{i=1}^t O(\mathbb{E}[|q(X_t) - q_\beta^*(X_t)|] + \mathbb{E}[(\widehat{r}_t(X_t) - r^*(X_t, \beta))^2]).$$

We defer the proof to Appendix A.3. The proof idea follows a similar idea that of the regret bound in Kato et al. [17] for deriving an estimator that is close to the optimal estimator for the average treatment effect in an adaptive experimentation setup. Theorem 3 relates the estimation error of $q_\beta^*(x)$ and $r^*(x, \beta)$ to how quickly $\beta$ will be deemed "safe". Hence, if we have good estimates of those quantities, then we can produce an estimates $(\widehat{\beta}_t)$ that are small and close to $\beta^*$ while remaining safe. We will now describe some practical methods for calculating $q_t$ and $\widehat{r}_t$, and demonstrate their empirical performance in some experiments.

## 4 Experiments

We use PyTorch to model our $(q_t)$ and $(\widehat{r}_t)$, and we consider the following formulations.[2]

1. **Baseline labeling policies.** We have baseline labeling policies of labeling all data that arrives, and a policy that just randomly samples $B$ proportion of samples to label — these are denoted respectively as "all" and "oblivious".

---

[2]Code at `github.com/neilzxu/active-rcps`

2. **Pretrain**: We derive an estimate of $r^*$ from a pretrained machine learning model, $\widehat{r}^{\text{pretr}}$, to be our choice of predictor for all time steps. We also derive an estimate of $\sigma(x, \beta)$, $\widehat{\sigma}^{\text{pretr}}(x, \beta)$, from the pretrained model. We also learn a sequence of normalizing constants $(C_t)$ s.t. the budget is satisfied. Our labeling policy in this case is $q_t(x) = \widehat{\sigma}(x, \widehat{\beta}_{t-1})/C_t$, where we want to optimize our policy for the previous best bound on $\beta^*$, $\widehat{\beta}_{t-1}$. We denote this method as "pretrain".

3. **Estimating $q_\beta^*$ and $r^*$**: We learn sequences of models $(\widehat{\sigma}_t^{\text{plugin}})$ and $(\widehat{r}_t^{\text{plugin}})$ using the labeled data points. We preprocess the outputs from $\widehat{\sigma}^{\text{pretr}}$ and $\widehat{r}^{\text{pretr}}$ to use as the input features to these models, respectively. Each of these sequences of models are then updated at every step. We also similarly learn a sequence of normalization constants $(C_t)$ for deriving the final labeling policy $(q_t)$.

We provide more details on how are methods are formulated in Appendix B. We run all our experiments on a 48-core CPU on the Azure platform, after using a GPU to precompute the predictions made by neural network models. We set $\theta = 0.1$, $\alpha = 0.05$, and $B = 0.3$ for all our experiments.

### 4.1 Numerical simulations

We have a simple data generating process of sampling $P_t \sim \text{Uniform}[0, 1]$ and let $Y_t \mid X_t \sim \text{Bern}(X_t)$. This simulates the setting we have with our real data where we have an accurate pretrained classifier that have a probability estimate of $Y_t$ of being 0 or 1. We let our covariates $X_t = P_t$. As a result, our risk function is the false positive rate $r_{\text{FPR}}(X, Y, \beta) := \mathbf{1}\{X \geq \beta, Y = 0\}$. We run 100 trials where each trial runs until 2500 labels are queried. We compare our methods based on their label efficiency, i.e., how close is $\widehat{\beta}_t$ to $\beta^* = 1 - \sqrt{2\alpha}$ after a set number of queried labels. In Figure 2, we plot the average $\widehat{\beta}_t$ reached after a given number of labels queried across trials. The shaded areas denotes pointwise 95% confidence intervals on the uncertainty of the average estimate. We can see that the "pretrain" and "learned" methods outperform both the "all" and "oblivious" strategies uniformly numbers across labels queried. In Figure 2a, we show the average rate of safety violations, i.e., the average proportion of trials that $\widehat{\beta}_t$ was unsafe and $\rho(\widehat{\beta}_t) > \theta$ at any time step. We can see that all methods control the desired safety violation rate at the predetermined level $\alpha$.

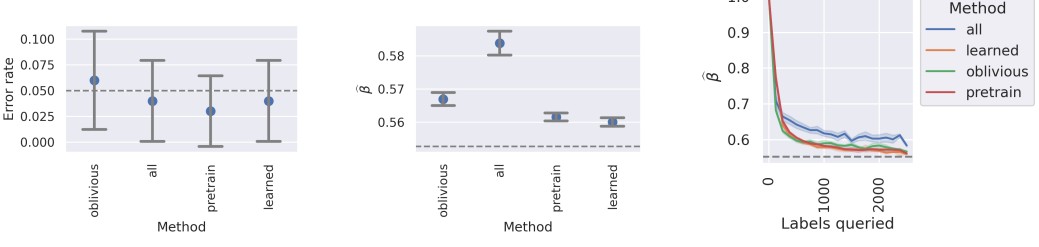

(a) Average rate of safety violations $\widehat{\beta}_t$.

(b) Average final value of $\widehat{\beta}_t$ (lower is better).

(c) Average $\widehat{\beta}_t$ vs. labels queried (lower is better).

Figure 2: Experimental results for different methods for our numerical simulation setup. We can see that "pretrain" and "learned" perform better by getting lower average $\widehat{\beta}_t$ uniformly across number of labels queried — the dotted line in Figures 2b and 2c is $\beta^* = 0.5578$. Each method also has low safety violation rate, i.e., is below the dotted line of $\alpha = 0.05$ in Figure 2a.

### 4.2 Imagenet

We also evaluate our methods on the Imagenet dataset [9], and we used the pretrained neural network classifiers from Bates et al. [4] to provide estimates of the class probabilities.

Since Imagenet is a classification task with label support on $\mathcal{Y} = [1000]$, our goal is to ensure that the miscoverage rate of the true class is controlled. We follow the same setup as descibed in the introduction, i.e., with our risk measure $r$ specified according to (1). For Imagenet, we reshuffle our dataset for each trial, and run each method till we have queried 3000 labels.

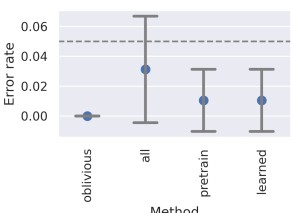 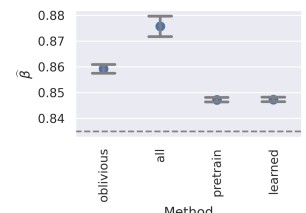 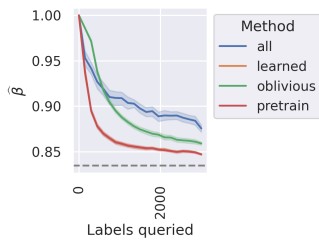

(a) Average rate of safety viola-
tions $\widehat{\beta}_t$.

(b) Average final value of $\widehat{\beta}_t$
(lower is better).

(c) Average $\widehat{\beta}_t$ vs. labels queried
(lower is better).

Figure 3: Experimental results for different methods on Imagenet. Again, we see that "pretrain"
and "learned" are the best performing, and they have very similar performance and hence overlap in
Figure 3c. Here, $\beta^* = 0.8349$, and is delineated by the dotted line in Figures 3b and 3c. Again, each
method also has low safety violation rate, i.e., is below the dotted line of $\alpha = 0.05$ in Figure 3a.

In Figure 3, we plot the average $\widehat{\beta}_t$ across trials. Once again, we can see that the "pretrain" and
"learned" methods outperform both the "all" and "oblivious" strategies here as well. On Imagenet the
average safety violation rate is also controlled as well under the predetermined level of $\alpha = 0.05$.

## 5 Additional related work

Casgrain et al. [6] provide anytime-valid sequential tests for identifiable functions, which result in
similar hypotheses being tested as this paper albeit with equality instead of equality. They, in addition
to other recent work [25, 24, 31], using regret bounds for betting-based e-processes to show either
derivations for the growth rate of a betting strategy w.r.t. to the optimal growth rate. However, none of
these settings incorporate the ability to perform adaptive sampling or inverse propensity weights. Prior
work in anytime-valid inference have included inverse propensity weights have been for off policy
evaluation [35], adaptive experimentation [7], or estimating the weighted mean of a finite population
[32]. However, none of these works explicitly characterize deviation in the sampling policy away
from the optimal sampling policy ultimately affects the growth rate as we do in Theorem 3.

Our analysis of power and regret for our algorithm is quite similar to methods in adaptive experimen-
tation for average treatment effect estimation [14, 17] that attempt to derive a no regret treatment
policy and outcome regressor that produces an estimator with a variance that approaches the variance
of the optimal estimator. Unlike the adaptive experimentation setting, however, we have an additional
label budget constraint on our formulation that results in a different optimal policy.

## 6 Conclusion, limitations, and future work

We have shown that we can extend the RCPS formulation to be anytime-valid, and retain validity and
increase label efficiency in an active learning setting. We use the theory of betting and e-processes
to develop this framework and show it is verifiably safe, and we verified this with our experimental
results. We have primarily considered the i.i.d. setting here for anytime-valid calibration, and one
key area in which one can extend this line of work is to account for distribution shift during test
time. The empirical Bernstein supermartingales in Waudby-Smith et al. [35] can likely be used to
extend our framework control risk in an average sense, but stronger guarantees could be made about
the provided risk control if more realistic assumptions are made about the nature of the distribution
(e.g., covariate shift, label shift, etc). It may also be possible extend a notion of adaptive conformal
inference (ACI) [11, 12] to anytime-valid risk control. Another limitation of this work is the bounded
label policy assumption (i.e., Assumption 2) and existence assumption in Proposition 5. We believe
that more careful analysis can get rid of these assumptions in future work.

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

# A Omitted proofs

Proofs that have been deferred from the main body of the paper are contained here.

## A.1 Proof of Proposition 4

We can rewrite the objective in the following way:

$$\mathbb{E}[\widehat{G}(q,\widehat{r},\lambda)] = \mathbb{E}\left[\lambda\left(\theta - \widehat{r}(X,\beta) - \frac{L}{q(X)}\bar{R}(\beta)\right) - \lambda^2\left(\theta - \widehat{r}(X,\beta) - \frac{L}{q(X)}\bar{R}(\beta)\right)^2\right]$$

$$= \lambda\left(\theta - \rho(\beta)\right) - \lambda^2\left((\theta - \rho(\beta))^2 + \mathbb{V}\left[\widehat{r}(X,\beta) + \frac{L}{q(X)}\bar{R}(\beta)\right]\right) \tag{11}$$

Maximizing (11) is the same as minimizing the following equivalent expressions:

$$\mathbb{V}\left[\widehat{r}(X,\beta) + \frac{L}{q(X)}\bar{R}(\beta)\right]$$

$$= \mathbb{E}\left[\mathbb{V}\left[\widehat{r}(X,\beta) + \frac{L}{q(X)}\bar{R}(\beta) \mid X\right]\right] + \mathbb{V}\left[\mathbb{E}\left[\widehat{r}(X,\beta) + \frac{L}{q(X)}\bar{R}(\beta) \mid X\right]\right]$$

$$= \mathbb{E}\left[\mathbb{V}\left[\frac{L}{q(X)}\bar{R}(\beta) \mid X\right]\right] + \mathbb{V}[R(\beta)]$$

$$= \mathbb{E}\left[\mathbb{E}\left[\frac{\bar{R}(\beta)^2}{q(X)} \mid X\right] - \mathbb{E}[\bar{R}(\beta) \mid X]^2\right] + \mathbb{V}[R(\beta)]$$

$$= \left(\int\left(\frac{\mathbb{E}\left[\bar{R}(\beta)^2 \mid X = x\right]}{q(x)} - \mathbb{E}[\bar{R}(\beta) \mid X = x]^2\right)p(x)\,dx\right) + \mathbb{V}[R(\beta)], \tag{12}$$

where we derive the 1st equality from the law of total variance, and the 2nd equality from the fact that $\widehat{r}(X,\beta)$ is fixed given $X$.

Since the only term that $\widehat{r}$ affects is the integral term in (12), we can choose each $\widehat{r}(x,\beta)$ for each $x \in \mathcal{X}$ to minimize the following:

$$\frac{\mathbb{E}\left[\bar{R}(\beta)^2 \mid X = x\right]}{q(x)} - \mathbb{E}[\bar{R}(\beta) \mid X = x]^2$$

$$= \frac{\mathbb{E}[(r(X,Y,\beta) - \widehat{r}(x,\beta))^2 \mid X = x]}{q(x)} - (\mathbb{E}[r(X,Y,\beta) \mid X = x] - \widehat{r}(x,\beta))^2$$

$$= \frac{\mathbb{E}[r(X,Y,\beta)^2 \mid X = x]}{q(x)} - \mathbb{E}[r(X,Y,\beta) \mid X = x]^2$$

$$\quad - \left(\frac{1}{q(x)} - 1\right)(2\mathbb{E}[r(X,Y,\beta) \mid X = x]\widehat{r}(x,\beta) - \widehat{r}(x,\beta)^2) \tag{13}$$

If we remove the constants (i.e., terms unaffected by $\widehat{r}(x,\beta)$), and note that $q(x)^{-1} - 1 > 0$, we get that minimizing (13) is equivalent to minimizing

$$2\mathbb{E}[r(X,Y,\beta) \mid X = x]\widehat{r}(x,\beta) - \widehat{r}(x,\beta)^2.$$

This is equivalent to minimizng the squared error, i.e., $(\mathbb{E}[r(X,Y,\beta) \mid X = x] - \widehat{r}(x,\beta))^2$, which means that $r^*(x,\beta) = \mathbb{E}[r(X,Y,\beta) \mid X = x]$, which gets us our desired result.

## A.2 Proof of Proposition 5

Since we have shown that maximizing (10) is equivalent to minimizing (12), we can isolate the terms that change wiht $q$ and see that we are looking for the solution to the following optimization problem:

$$\min_q \int \frac{p(x)}{q(x)}\mathbb{E}[\bar{R}(\beta)^2 \mid X = x]\,dx$$

$$\text{s.t.} \int p(x)q(x)\,dx \leq B.$$

We can define $\varphi(x) := p(x)q(x)$ rewrite this as

$$\min_{\varphi} \int \frac{p(x)^2}{\varphi(x)} \mathbb{E}[\bar{R}(\beta)^2 \mid X_i = x] \, dx$$

$$\text{s.t.} \int \varphi(x) \, dx \leq B.$$

Assume we can define a valid $q_\beta^*$ where $q_\beta^*(x) \in [0, 1]$ for each $x \in \mathcal{X}$ that satisfies the following conditions:

$$\varphi(x) \propto p(x)\sqrt{\mathbb{E}\left[\bar{R}(\beta)^2 \mid X = x\right]}$$

$$q_\beta^*(x) \propto \sqrt{\mathbb{E}\left[\bar{R}(\beta)^2 \mid X = x\right]}.$$

Explicitly, we define $q_\beta^*$ as follows:

$$q_\beta^*(x) = \frac{\sqrt{\mathbb{E}\left[\bar{R}(\beta)^2 \mid X = x\right]}}{\mathbb{E}\left[\sqrt{\mathbb{E}\left[\bar{R}(\beta)^2 \mid X = x\right]}\right]} \cdot B.$$

One can show its optimality by considering some other labeling policy $q'$ where $\mathbb{E}[q'(X)] = B$ (we use a similar proof technique from importance sampling Owen [23, § 9.1]). Now, let $\varphi^*(x) := p(x)q_\beta^*(x)$ and $\varphi'(x) = p(x)q'(x)$

$$\int \frac{p(x)^2}{\varphi^*(x)} \mathbb{E}[\bar{R}(\beta)^2 \mid X_i = x] \, dx$$

$$= \mathbb{E}\left[\sqrt{\mathbb{E}\left[\bar{R}(\beta)^2 \mid X = x\right]}\right] \int \frac{p(x)}{B} \cdot \sqrt{\mathbb{E}[\bar{R}(\beta)^2 \mid X_i = x]} \, dx$$

$$= \mathbb{E}\left[\sqrt{\mathbb{E}\left[\bar{R}(\beta)^2 \mid X = x\right]}\right]^2 \cdot B^{-1}$$

$$= \left(\int \frac{p(x)}{\varphi'(x)} \cdot \varphi'(x) \cdot \sqrt{\mathbb{E}\left[\bar{R}(\beta)^2 \mid X = x\right]} \, dx\right)^2 \cdot B^{-1}$$

$$= B \cdot \left(\int \frac{p(x)}{\varphi'(x)} \cdot \frac{\varphi'(x)}{B} \cdot \sqrt{\mathbb{E}\left[\bar{R}(\beta)^2 \mid X = x\right]} \, dx\right)^2$$

$$\leq \int \frac{p(x)^2}{\varphi'(x)} \mathbb{E}\left[\bar{R}(\beta)^2 \mid X = x\right] \, dx,$$

where the last line is by Cauchy-Schwarz, since $\varphi'(x)/B$ is a valid p.d.f. Hence, we have shown our desired result.

### A.3   Proof of Theorem 3

By definition of $\text{Reg}_t$, we know that

$$\sum_{i=1}^{t} \mathbb{E}[\widehat{G}^\beta(q_i, \widehat{r}_i, \lambda^*)] - \mathbb{E}[\widehat{G}^\beta(q_t, \widehat{r}_i, \lambda_i)] \leq \text{Reg}_t$$

by taking an expectation over $\mathcal{F}_{i-1}$ for each term in the summation Hence, what remains to be shown is the following:

$$\sum_{i=1}^{t} \mathbb{E}[\widehat{G}^\beta(q^*, r^*, \lambda^*)] - \mathbb{E}[\widehat{G}^\beta(q_t, \widehat{r}_t, \lambda^*)]$$

$$\leq \sum_{i=1}^{t} O(\mathbb{E}[|q_t(X_t) - q_\beta^*(X_t)|]) + O(\mathbb{E}[(\widehat{r}_t(X_t, \beta) - r^*(X_t, \beta))^2]) \qquad (14)$$

Let $\bar{R}_t(\beta) := r(X, Y, \beta) - \widehat{r}_t(X, \beta)$ and $\bar{R}^*(\beta) := r(X, Y, \beta) - r^*(X, \beta)$. We first note the following identity using (12):

$$\mathbb{E}[\widehat{G}^\beta(q^*, r^*, \lambda^*)] - \mathbb{E}[\widehat{G}^\beta(q_t, \widehat{r}_t, \lambda^*)]$$

$$= (\lambda^*)^2 \left( \mathbb{E}\left[ \mathbb{V}\left[ \widehat{r}_t(X, \beta) - \frac{L}{q_t(X)} \bar{R}_t(\beta) \mid \mathcal{F}_{t-1} \right] - \mathbb{V}\left[ r^*(X, \beta) - \frac{L}{q^*_\beta(X)} \bar{R}^*(\beta) \right] \right] \right)$$

Now, we make the following derivations for the difference between the variance ($\mathbb{V}$) terms:

$$\mathbb{V}\left[ \widehat{r}_t(X, \beta) - \frac{L}{q_t(X)} \bar{R}_t(\beta) \mid \mathcal{F}_{t-1} \right] - \mathbb{V}\left[ r^*(X, \beta) - \frac{L}{q^*_\beta(X)} \bar{R}^*(\beta) \right]$$

$$= \int \left( \frac{\mathbb{E}[\bar{R}_t(\beta)^2 \mid X = x]}{q_t(x)} - \frac{\mathbb{E}[\bar{R}^*(\beta)^2 \mid X = x]}{q^*_\beta(x)} - \mathbb{E}[\bar{R}_t(\beta) \mid X = x]^2 \right) \cdot p(x) \, dx$$

$$= \int \left( \frac{(q^*_\beta(x) - q_t(x))\mathbb{V}[r(X, Y, \beta) \mid X = x] + q^*_\beta(x)(1 - q_t(x))(\widehat{r}_t(x) - r^*(x, \beta))^2}{q_t(x)q^*_\beta(x)} \right) \cdot p(x) \, dx$$

$$\leq \int \left( (q^*_\beta(x) - q_t(x)) + (\widehat{r}_t(x) - r^*(x, \beta))^2 \right) \cdot \frac{p(x)}{\varepsilon} \, dx$$

$$\leq O(\mathbb{E}[|q^*_\beta(X_t) - q_t(X_t)| \mid \mathcal{F}_{t-1}] + \mathbb{E}[(\widehat{r}_t(X_t) - r^*(X_t, \beta))^2 \mid \mathcal{F}_{t-1}]). \tag{15}$$

The 1st equality is by substituting in the identity from (12). The 1st inequality is a result of $\mathbb{V}[r(X, Y, \beta) \mid X = x] \leq \frac{1}{4}$, since $r(X, Y, \beta) \in [0, 1]$, and $q^*_\beta(x), q_t(x) \in [\varepsilon, 1]$ almost surely. The 2nd inequality is by upper bounding $q^*_\beta(x) - q_t(x)$ by its absolute value.

Now, if we plug (15) into (14), take the expectation over $\mathcal{F}_{t-1}$, and take the summation over $t$, we get our desired result.

# B    Experiment details

In this section, we discuss additional details about how we implement our methods described in Section 4.

### B.1    Formulation of the labeling policy

For the "pretrain" policy, we use an estimate of the conditional mean and variance derived from $s$.

$$\widehat{r}^{\text{pretr}}(x, \beta) := \sum_{y \notin C(x, \beta)} s^y(x),$$

$$\widehat{\sigma}^{\text{pretr}}(x, \beta) := \sqrt{\widehat{r}^{\text{pretr}}(x, \beta) \cdot (1 - \widehat{r}^{\text{pretr}}(x, \beta))}.$$

These estimates may not be accurate, but might still represent a reasonable partitioning of the feature space where $\sigma(x, \beta)$ and $r^*(x, \beta)$ are similar. Hence, for $\widehat{\sigma}^{\text{plugin}}_t$ and $\widehat{r}^{\text{plugin}}_t$, we model them as linear regresssion models where inputs are a binning of $\widehat{r}^{\text{pretr}}(x, \beta)$ and $\widehat{\sigma}^{\text{pretr}}(x, \beta)$, respectively. We then learn the regression model parameters on training data.

### B.2    Optimization to maintain the budget constraint

For any predictor $\widehat{\sigma}$, we optimize the Lagrangian corresponding to (10), which is defined as follows for a fixed $\lambda_t$.

$$\mathcal{L}(\nu_t, q_t)$$

$$= \mathbb{E}[\widehat{G}(q_t, \widehat{r}_t, \lambda_t)] - \nu_t(\mathbb{E}[q_t(X_t)] - B)$$

$$= \mathbb{E}\left[\lambda_t\left(\theta - \frac{L_t}{q_t(X)}r(X,Y,\beta)\right) - \psi(\lambda_t)\left(\theta - \frac{L_t}{q_t(X)}r(X,Y,\beta)\right)^2\right] - \nu_t(\mathbb{E}[q_t(X)] - B)$$

$$= \lambda_t(\theta - \rho(\beta)) - \lambda_t^2\mathbb{E}\left[\left(\theta - \widehat{r}_t(X,\beta) - \frac{L_t}{q_t(X)}\bar{R}(\beta)\right)^2\right] - \nu_t(\mathbb{E}[q_t(X)] - B)$$

Since we know the optimal form of $\widehat{r}_t$, we optimize it separately by taking an optimization step with the loss of squared error, $(r(X,Y,\beta) - \widehat{r}_t(X,\beta))^2$, for each labeled example for a grid of $\beta$ values.

To derive the solution, we simplify playing the minimax game with the above Lagrangian to the following objective:

$$\max_{q_t}\min_{\nu_t} - \mathbb{E}\left[\frac{1}{q_t(X)} \cdot (r(X,Y,\beta) - \widehat{r}_t(X,\beta))^2\right] - \nu_t(\mathbb{E}[q_t(X)] - B)$$

In the case of both "pretrain" and "learned" methods, we parameterize our $q_t$ in the following fashion:

$$q_t(x) = \frac{\widehat{\sigma}_t(x, \widehat{\beta}_{t-1})}{\exp(c_t)},$$

where our normalization constant $C_t = \exp(c_t)$ for some value $c_t \in \mathbb{R}$ to ensure it is nonnegative.

$\widehat{\sigma}_t$ is updated separately. For "pretrain", it is fixed from the beginning, and for "learned", we take an optimization step to minimize the squared loss against the squared residual, i.e., we update to minimize $((r(X,Y,\beta) - \widehat{r}_t(X,\beta))^2 - \widehat{\sigma}_t(X,\beta)^2)^2$.

Hence, the only thing that remains to optimize $c_t$, which now simply means we need to solve the following problem (where we treat $\widehat{\beta}_{t-1}$ as fixed):

$$\max_{c_t}\min_{\nu_t} - c_t + \nu_t\left(\frac{1}{\exp(c_t)}\mathbb{E}[\widehat{\sigma}_t(X,\widehat{\beta}_{t-1})] - B\right)$$

The actual game payoff we play is the stochastic approximation of the Lagrangian in the following form:

$$L(\nu_t, c_t) = -c_t - \nu_t\left(\frac{\widehat{\sigma}_t(X_t, \widehat{\beta}_{t-1})}{\exp(c_t)} - B\right).$$

At each step, we take an optimization step on $c_t$ towards maximizing the above loss, and determine $\nu_t$ by playing either best response or a windowed best response that takes an average of best responses over recent rounds. We use the COCOB optimizer [22] for all of learning and optimization which requires no hyperparameter or learning rate selection.

