# OpenReview forum: "Active, anytime-valid risk controlling prediction sets"
_NeurIPS.cc/2024/Conference — NeurIPS 2024 poster_

### Official Review · Reviewer_XpAy · 2024-07-09

**Soundness:** 4
**Presentation:** 2
**Contribution:** 2
**Rating:** 6
**Confidence:** 2

**Summary:**

This paper proposes a probabilistic strategy for stream-based active learning which allows for construction of anytime valid prediction sets. The strategy stems from maximizing the variance process of an e-process, which is also used in designing the prediction sets. Relevant theoretical guarantees (validity and regret) and proof-of-concept experiments are presented.

**Strengths:**

For me, the most interesting contribution of the paper, is that it brings the perspective of safe betting to active learning.
The results are not written in the most flashy way, but the overall thought process is straightforward to follow and proofs are clear.

**Weaknesses:**

- My assessment is that Section 2 lacks novelty, as in, very similar results have been proposed in earlier work under slightly different problem setting. Would be nice to make section 3 the main contribution of the paper, and present section 2 as a background/recipe section which leads to results of Section 3.
  - The proposed e-processes of Eq (5) and (6) are both from results of Waudby-Smith [2023 & 2024]. If i understand correctly, both of these processes are proposed for a more general frame-work, and the current paper applies it to its problem setting.
  - Proposition 1 and Theorem 1 are immediate corollaries by definition of the betting e-processes.
    - It is best that this is merged together and presented as a corollary for the specific choice of $M_t$.
    - Theorem 1 holds true for any $M_t$ is an e-process under the null. In the proof however, Proposition 1 is invoked. Would be good if the statement is updated and the assumption on $M_t$ is stated in it.

- I am not widely aware of the literature. I just wanted to raise this flag that the related works section might be inadequate. I was expecting more work to have been done in this area. After reading the paper, I do not have a clear image of short history of results leading to this work, and I think this speaks of an incomplete literature review. There's no related work from active learning literature, while they aim to solve a relatively close problem (in cases identical).


- The contributions section, particularly the second paragraph, is too technical given what it is already introduced. I would generally shorten the introduction and present a more thorough literature review in the main text.
- I think parts of the problem setting are confusing, if not incorrect and lines 44-52 should be re-written.
  - Line 47 says $f$ outputs and action $A$ within a set $\mathcal A$. But later in Line 50, it is written that $f$ is a prediction set for $Y$.
  - Connection between $\mathcal A$ and $\mathcal Y$ is missing. I'm assuming $f$ output subsets of $\mathcal Y$ and I don't understand the auxiliary confusing notation of $A$ and $\mathcal A$.
  - If $f$ outputs a prediction set, then the notation $A$ and calling it an action is utterly confusing. Within active learning literature, $X$ is typically the action.
  - It is not evident in the notation of the risk that $\rho$ depends on $f$.

**Questions:**

1) Why is assumption 1 in terms of $\rho$ and not just $r$? Is this needed? If possible, stating the assumption in terms of $r$ is cleaner in my opinion, since this way $\mathcal X$ and $\mathcal Y$ become assumption free.
2) The way section 2.2 is stated, what I collect from it is that the $M_t$ process is robust to adding a $\mathcal F_t$-adapted and bounded bias term. I can imagine that adding this bias may help with rapid increase of variance process of $M_t$. But I did not understand the motivation of this variance reduction, within the context of section 2. Could you please elaborate?
3) The proposed optimal labeling policy in Section 3 somewhat resembles a variance maximization techniques which are standard practice in stream-based active learning. Have you looked into the connections? How do these relate?
4) Doesn't the problem setting allow for comparison to standard active learning policies? Adding these would make the paper relevant to a considerable larger community.

**Limitations:**

Limitations and assumptions are adequately stated.

---

> ### Author Rebuttal · Authors · 2024-08-07
>
> Our responses to the highlighted weaknesses are as follows.
>
> - **Novelty of e-processes.** Thank you for pointing out the error in the proof of Theorem 1 --- we will correct as you have described. While we agree that betting e-process was introduced in [1] and an inverse propensity weighted form was proposed in [2], we disagree that is simply an application of existing work. Our methods are novel in several ways.
>     1. The betting processes of [1, 2] are used to derive confidence sequences for the mean of a bounded (or lower bounded with bounded mean) random variable. In our setting, each betting process does not assume there is a different mean of the random variable tested under the null --- it assumes the same lower bound on the mean, that is $E[r(X, Y, \beta)] \geq \theta$, for each value of $\beta$. Here, our goal is to estimate $\beta^*$ --- we only use the fact that under the null, we can derive a random variable that has a bounded mean, and has a lower bound. Our methods are more similar to those in Lemmas 3.1 and 3.2 in [3]. In that work, however, the authors are only interested in designing confidence sequences for estimation, rather than outputting a parameter with risk control.
>     2. The anytime-valid risk control guarantee in Definition 1 is not the same as a merely providing a confidence sequence (CS) for $\beta^\*$. If one only provided a confidence sequence, it would not be clear which $\beta$ one should choose in the CS to ensure that $\rho(\beta) \leq \theta$ --- one of them is $\beta^*$ with high probability, but no guarantees would be made about any of the risk of any other $\beta$, we would not be able to guarantee we can always output a $\hat{\beta}_t$ that has risk control at each time step. Thus, key to showing the risk control guarantee in Definition 1 is also the assumption that $\rho$ is monotonic in $\beta$. The choice of $\beta$ that are safe and provide risk control are the ones that have been eliminated from the CS. In fact, the null of our e-process is *the opposite* of the risk control we want to achieve, since our resulting CS captures the set of $\beta$ that still might have risk that is *at least* $\beta$. This is a relatively novel formulation. Creating a CS to capture the opposite of the desired result one would want probabilistic guarantees for (in some sense) has been used in auditing election results with risk-limiting audits [4], but the guarantees and use in that application are quite different than ours.
>     3. In [2], there is an assumption that one always has access to an outcome $Y$ for each time step(i.e., the reward), since the application of interest is off policy evaluation, and some action is taken at each time step, and the corresponding reward is obtained. In our setting, we do not receive the outcome $Y$ at all if we did not query a label --- however, the weighted estimator is an unbiased estimator of the actual $r(X, Y, \beta)$ incurred at each time step, so our e-process is still correct.
> - **Relate work on active learning.** See our point on comparing with active learning in the top-level rebuttal.
> - **Problem setting seems confusing.** In line 50, we say that our action is a prediction set for $Y$ in many applications, b/c the risk we wish to control miscoverage. However, the action doesn't need to be a prediction set (e.g., could be the behavior policy of a safe robot [5]). Our usage of $X$ and $Y$ as covariate and label is standard from previous papers on RCPS, but we will make clear what $A$ means and refers to in our final version. We will also clarify the relationship between $\rho$ and $f$ explicitly. The relationship is as follows: we define $\rho(\beta) = E[r(X, Y, \beta)] = E[\ell(Y, f(X, \beta))]$.
>
> Our responses to the questions are as follows.
>
> 1. **Assumption of monotonicity on $\rho$.** The assumption on $\rho$ is slightly more general than assuming $r$ is monotonic directly. We appreciate the suggestion and will clarify that monotonicity of $r$ implies monotonicity of $\rho$ and this is a useful special case of our assumption that is distribution free.
> 2. **How the predictor reduces variance.** The e-process in section 2.2 has $\bar{R}(\beta)$ inversely weighted by the probability of labeling when a label is queried. We can see that without the predictor set to always output 0, the variance of $\hat{r}(X, \beta) + L / q(X) * \bar{R}(\beta)$ is larger than $r(X, Y, \beta)$. However, in the extreme, if we had a perfect predictor, i.e., $r^*(X, \beta) = r(X, Y, \beta)$, then we recover exactly $r(X, Y, \beta)$. This has lower variance, and would result in our growth rate being identical to the e-process in Section 2 that queries every label. For more formal analysis, Theorem 3 encapsulates how the accuracy of the predictor results in changes in the growth rate of our e-processes.
> 3. **Relationship to variance maximization in active learning.** Our optimal policy aims to sample covariates $X$ with probability proportional to square root of the expected conditional risk squared --- when the optimal predictor is used, this quantity is the same as the conditional standard deviation of the risk. This is similar in spirit to existing active learning algorithms that aim to select data points that have the highest variance of predictions (e.g., [6]). There is a subtle point here that variance in predictions is not precisely the same as the variance in risk (i.e., in a prediction set setting, if the different predictions are covered by the prediction set anyways, then there is no variance in risk). In addition, many active learning algorithms are deterministic, i.e., they pick the points that best fit a criterion(e.g., conditional variance, disagreement, diversity, etc.) rather than sampling them with some probability.
> 4. **Comparison to standard active learning policies.** See previous points on related work in active learning and the relationship to variance maximization.

---

> ### Author Response · Authors · 2024-08-07
> **Rebuttal (continued)**
>
> **Additional comments on problem setting**. For example, we may wish to calibrate the aggressiveness of a robot's behavior policy that is parameterized by $\beta$ (e.g., [5]), where the goal is to reach a destination while avoiding obstacles. The action $A$ may be the control policy of the robot based on the environment, and the risk would be the distance to the nearest obstacle over the entire trajectory of the robot as it travels to its destination. We may want this risk to be controlled on average over the distribution of environments. Our framework applies to such as setup, and hence we term the output based on the covariate $X$ an action $A = f(X)$ rather than limiting it to only being a prediction set.
>
> We also include our top level comment on related work to active learning here:
>
> **Comparison to active learning.** We briefly summarize the differences and similarities between active learning and our problem. Our problem objective and the methods use to prove their validity are different from typical active learning methods. Active learning (stream-based and pool-based) aims to minimize label queries and still learn the best possible machine learning predictor. Guarantees in this area usually are model-based or learning theoretic, i.e., they propose a model update/selection procedure and query procedure that minimizes the true risk of the model over a known or arbitrary function class, and derive results using a notion of class complexity (when one is developing a procedure that agnostic to the exact function class), or the methods are evaluated empirically for specific models, without guarantees. In contrast, our procedure tunes a calibration parameter that can be wrapped around any black-box model to provide a statistically rigorous risk guarantee. As a result, it means that other types of querying strategies that are deterministic (e.g., disagreement based, diversity based, etc.) cannot be directly imported to our problem setting, since the statistical guarantees we derive require that our queries are probabilistic. Further, we do not think existing active learning necessarily tackle the same objective, since they focus primarily on optimizing the performance of a classifier, rather than guaranteeing risk control while calibrating a parameter. Further development of how to leverage active learning methods in our setting is a fruitful direction for future work.
>
> **References**
>
> [1] I. Waudby-Smith and A. Ramdas. Estimating means of bounded random variables by betting. *Journal of the Royal Statistical Society Series B (Statistical Methodology)*, 2023.
>
> [2] I. Waudby-Smith, L. Wu, A. Ramdas, N. Karampatziakis, and P. Mineiro. Anytime-valid off-policy inference for contextual bandits. *ACM / IMS Journal of Data Science*, 2024.
>
> [3] P. Casgrain, M. Larsson, and J. Ziegel. Sequential testing for elicitable functionals via supermartingales. *Bernoulli*, 2024.
>
> [4] I. Waudby-Smith, P. B. Stark, and A. Ramdas. Rilacs: Risk limiting audits via confidence sequences. *International Joint Conference on Electronic Voting*, 2021.
>
> [5] J. Lekeufack, A. N. Angelopoulos, A. Bajcsy, M. I. Jordan, and J. Malik. Conformal Decision Theory: Safe Autonomous Decisions from Imperfect Predictions. arXiv:2310.05921, 2024.
>
> [6] D. Cacciarelli and M. Kulahci. Active learning for data streams: a survey. *Machine Learning*, 2024.

---

> > ### Comment · Reviewer_XpAy · 2024-08-13
> >
> > Thank you for your answers, and I apologize for the late response.
> >
> > My questions are addressed, and I hope that the paper is updated accordingly -- I still think that the contributions would benefit a lot from a more crisp presentation e.g. by positioning the work more vividly.
> >
> > Overall, I agree with reviewer 1rh6 and recommend for acceptance.

---

### Official Review · Reviewer_qKw8 · 2024-07-29

**Soundness:** 2
**Presentation:** 2
**Contribution:** 2
**Rating:** 4
**Confidence:** 1

**Summary:**

The authors, in their work, extend the framework of Risk Controlling Prediction Sets (RCPS) to a sequential setting where data is collected adaptively, providing anytime-valid risk guarantees. Additionally, it proposes a framework for active labeling, which allows for selective querying of true labels within a predefined budget, enhancing the utility of RCPS by leveraging predictors to estimate expected risk based on covariates. Next, the authors extend the setting further and develop an active learning setting and optimal labeling policy and under a fixed label budget.

**Strengths:**

- Extended setting of a risk-controlling prediction set setting.
- Possibly rigorous in theoretical analysis.

**Weaknesses:**

- The paper is notation-heavy, considers complex settings, and uses concepts that are not so commonly known. Unfortunately, the authors are doing nothing to help readers understand their work.
- Many symbols (and there are many) are only introduced once.
- The goal of the method is easy to miss in the body of the text.
- The experimental section is hard to understand without reading the theory sections, and there are no meaningful conclusions.
- There are no summaries, conclusions, schemas, pseudo-codes, or additional intuitions to help the reader understand the paper.
- Experiments feel very limited.
- I have doubts about the practicality of the introduced setting, as the paper needs better application examples.

Due to very limited time, I was not able to read related works that were necessary to fully understand this paper, verify the theory, and possibly appreciate it fully. I would like to believe that all the theories presented in the paper are right, but even assuming that. I think the paper is below acceptance due to quite a poor presentation.

**Questions:**

- What are other examples of applications for the proposed framework?
- How in practise one should select $\theta$ and $\alpha$?

**Limitations:**

I see no potential negative social impact of this work. Discussion on limitations is limited.

---

> ### Author Rebuttal · Authors · 2024-08-07
>
> Our response to each highlighted weakness and question are as follows.
> - **Notational usage and clearer introduction of concepts.** We will clean up our notational usage and introduce more unfamiliar concepts more comprehensively (e.g., risk controlling prediction sets, e-processes, anytime-validity, etc.) in our final draft.
> - **Symbol usage.** We will simplify this as well to make it more accessible, including incorporating the suggestions put forth by reviewer XpAy about the notation for the problem setting.
> - **Goal of method is easy to miss.** The goal of our method is specified in Definition 1 and (3) in our paper --- we will highlight this definition and make it more apparent in the final version.
> - **Experimental section requires reading theory section.** We will separate out the methods we are employing from the theoretical section so they can be read in a standalone fashion (i.e., in an algorithm environment). We do have meaningful conclusions concerning our experiments --- the "pretrain" and "learned" label policy/predictor combinations that both utilize the machine learning model being calibrated outperform (in terms of label querying efficiency) the naive baseline labeling policies/predictor combos. This shows that, in practice, the models we are calibrating can be used to estimate the conditional risk and variance to a sufficient degree of accuracy such that it significantly improves the label efficiency over the baseline approaches. We will clarify this emphasis in our final version.
> - **Limited experiments**: We will aim to add more experiments from different machine learning methods and risk functions to illustrate the efficacy of our procedure (i.e., QA task for LLMs, and image labeling using the COCO-MS dataset).
> - **Practicality of methods.** We will provide more concrete examples of applications, but we think our framework is quite widely applicable. Here are some examples of applications:
>     - *Reduce query cost in medical imaging.* A medical imaging system that outputs scores for each pixel of image that determines whether there is a lesion or not would want to utilize labels given by medical experts for unlabeled images from new patients. Since the cost of asking experts to label these images is quite high, one would want to query experts efficiently, and only on data that would be most helpful for reducing the number of highlighted pixels.
>     - *Domain adaptation for behavior prediction.* One reason we would want online calibration in a production setting is that we may have much different distribution of data that we do not have access to before deployment. For example, during a navigation task for a robot, we may want to predict the actions of other agents and avoid colliding into them when travelling between two points [1]. Since agents may behave differently in every environment, it makes sense to collect the behavior data in the test environment and update the behavior prediction in an online fashion to get accurate predictions calibrated for specifically the test environment.
>     - *Safe outputs for large language models (LLMs).* One of the goals with large language models is to ensure their responses are not harmful in some fashion (e.g., factually wrong, toxic, etc.). One can view this as outputting a prediction set for the binary label set of $Y \in \\{\texttt{harmful},\ \texttt{not harmful}\\}$. Many pipelines for modern LLMs include some form of a safety classifier, which scores the risk level of an output, and determines whether it should be output to the user or not [2, 3], or a default backup response should be used instead. One would want to label production data acquired from user interaction with the LLM and used to calibrate cutoff for the scores that are considered low enough for the response to be allowed through.
> - **Choice of $\theta$ and $\alpha$.** The choice of $\theta$ and $\alpha$ will depend on the application the user has in mind. Though, since $\alpha$ is determines the probability the bound holds, reasonable default is to choose it $\alpha=0.05$. We would like to reiterate that a particular utility of our method is that the probability bound holds *uniformly over every time step*, that is the probability that there is a single $\hat{\beta}_t$ for all $t \in \mathbb{N}$ that has risk greater than $\theta$ is less than $\alpha = 0.05$. $\theta$ should be chosen based on the risk metric being controlled, and a reasonable default choice could also to be choose $\theta=0.05$. In the context of image classification, this would mean that the prediction set of possible classes will not cover the true label only 5\% of the time on average. We will elucidate this point more in the final version.
>
>
> **References**
>
> [1] J. Lekeufack, A. N. Angelopoulos, A. Bajcsy, M. I. Jordan, and J. Malik. Conformal Decision Theory: Safe Autonomous Decisions from Imperfect Predictions. arXiv:2310.05921, 2024.
>
> [2] T. Markov, C. Zhang, S. Agarwal, F. E. Nekoul, T. Lee, S. Adler, A. Jiang, and L. Weng. A Holistic Approach to Undesired Content Detection in the Real World. *AAAI*, 2023
>
> [3] L. Hanu and Unitary team. Detoxify. Github. https://github.com/unitaryai/detoxify, 2020.

---

> > ### Comment · Reviewer_qKw8 · 2024-08-13
> > **Re: Rebuttal by Authors**
> >
> > Thank you for your detailed response to my rather short review. Other reviews also pointed out areas for improvement in terms of the presentation, so I hope you will address them in the next revision of your paper. Since the promised improvement in the presentation cannot be verified at the moment, I keep my score as it is. However, I'm okay with your work being accepted, since the other reviews recommended it.

---

### Official Review · Reviewer_1rh6 · 2024-07-29

**Soundness:** 4
**Presentation:** 2
**Contribution:** 3
**Rating:** 7
**Confidence:** 3

**Summary:**

The setting extends the model of Bates et al.---which provide confidence-bound type guarantees on the performance of a trained ``black-box" predictor which are parametrised, and nested with respect to a monotonic parameter $\beta$---to the online setting. The goal of the original setting is to provide risk-controlling prediction sets (RCPS) mapping a feature $X$ to a set $\mathcal{T}(X)$ in the label space, which is judged on the basis of a predetermined safety criterion. The online extension is then well-justified by the observation that the calibration data itself is often limited without deploying the model in practice, particularly if training occurs in an online fashion. \\
The contributions of the paper are:

1. An extension of the RCPS notion to the online setting, and a derivation of guarantees that are anytime-valid. In other words, confidence sets are refined as data is accrued, nevertheless maintaining risk control over the entire stream.
2. An extension of the RCPS which is valid under active learning, where a learner may choose to obtain a label based on the covariates, and a fixed total query budget.

In addition, the authors provide guarantees in the regret sense on the performance of derived methods in terms of the log-optimality criterion common for evaluating anytime-valid methods, which decouples into the regret of an exp-concave sequence dependent only on a quantity called the betting fraction, which due to log-concavity may be estimated sub-linearly in the number of rounds $T$ (due to exp-concavity of the log-optimality with respect to the betting fraction, this should be $O(\log(T))$) plus two concentration-type terms dependent on the convergence of two additional parameters derived from the risk, which may be estimated from the data. The convergence of such estimates will depend on the variance of the classifier's risk, and is determined on a case-by case basis. Experiments are included for verification of theoretical guarantees.


Conclusion:
While there are some weaknesses with regard to the communicability of the results for the intended audience, I believe on balance that the ideas in this paper will turn out to be of broad interest to the experimental community, and have clear practical relevance. From the theoretical perspective, although many of the ideas are directly generated from the theory of e-statistics, this is an encouraging example of tailoring theory to a clear practical need.

**Strengths:**

1. The formalism is well developed, and demonstrates a high-degree of understanding of the use of e-variables as diagnostic tools of testing processes, and the derived confidence methods are a natural extension of the theory to a very practical setting.
2. The extension to label-efficient active learning further pushes the abstract theory into the realms of practicability, and the authors provide an example of a set of estimators for the various relevant quantities in the experiments. This gives an (somewhat implicit) recipe for practitioners looking to deploy the tools developed.
3. The regret bound derived gives an interpretable decomposition of the optimality of the regret of the growth-function (used to bound the log-optimality criterion), which decouples statistical quantities from the betting fraction, which is obtainable from standard online learning methods.
4. The experiments (although arguably non-extensive) provide minimal necessary examples and concrete estimators for the relevant quantities present in the regret bounds, and demonstrate empirically the convergence of the sum total of relevant components to their optimal values.

**Weaknesses:**

1. A few more examples of the method being used in practice, along with the guarantees from the regret bound explicated, conditional upon the estimators idiosyncratic to the exact settings treated would be much more helpful than the abstract bound. Although for a theorist these results make sense, and keeping the general form due to the presence of the risk estimator variance and other empirical quantities does find some justification due to the variability of these quantities (which may alter the order of the regret bound), a few examples would really help illustrate the process + guarantees for a practitioner, which I guess would be the intended audience. I think this paper highlights the difficulty of the communicability of theoretical results in a digestible fashion, but does not constitute a weakness of the results themselves.
2. The experiments are not particularly extensive, and only encompass two examples; a contrived example of uniformly drawn features with Bernoulli labels (which is still a demonstrative diagnostic for the methods described, but have very well-behaved associated estimates), and a more realistic example based on the Imagenet dataset. Furthermore, as mentioned above, the comparison of online performance with respect to the regret bound would be even more helpful.
3. It would be nice to see some experiments illustrated with harder examples, for example in cases where the convergence of estimates would have a different than $O(\sqrt{T})$ rate.

**Questions:**

1. To my understanding, the convergence rate of the estimation terms largely dominate the $O(\log(T))$ rate from the $G^{\beta}$-contribution. Are there any non-trivial examples where this isn't the case?
2. Is there any hope of obtaining an adaptive rate in the growth function---perhaps by means of a tailored variance-reduced estimator---which could yield an instance-dependent result, if the true risk-variance is low? The estimation of $q$ and $r$ seems like it might be particularly troublesome if one tries to derive something for a general case, but this might be a misunderstanding on my part.

**Limitations:**

I have discussed limitations of the work above, and it seems that the authors have laid out the limitations of the degree of practicability due to the i.i.d. assumption (i.e., if training is coupled to the observed data stream). Nevertheless, I still think this is a solid contribution. With regard to the societal impact, the development of a robust theory of confidence in machine learning is undoubtedly a good thing.

---

> ### Author Rebuttal · Authors · 2024-08-07
>
> Here are our point-by-point responses to each of the highlighted weaknesses.
>
> 1. **More examples of the method being used in practice, along with examples of choice of estimators for the labeling policy and control variate regression.** We agree that providing more examples would illustrate the utility of our method better, and will add more concrete examples to our final draft. In particular, we note that often machine learning models have some estimate $\hat{P}(X)$ of the conditional distribution of $Y \mid X$ (e.g, class probabilities, conditional diffusion models, LLMs, etc.). Thus, for any realized covariate $x$,  we can derive use $\mathbb{E}_{Y \sim \hat{P}(x)}[r(X, Y, \beta) \mid X = x]$ from the machine learning model as our choice of $\widehat{r}(x)$. This expectation can either be calculated analytically (as we do in or classification examples in our experiments) or derived using Monte Carlo (for generative models/LLMs where one can sample from the conditional distribution). In essence, we are already getting a predictor (for free in some sense) from the very model we are calibrating.
> 2. **More extensive experiments and empirical analysis of regret bound**. We will aim to add more experiments from different machine learning methods and risk functions to illustrate the efficacy of our procedure (i.e., QA task for LLMs, and image labeling using the COCO-MS dataset), as well as an empirical analysis of our regret bound in our experiments.
> 3. **Experiments with estimators of varying rates of convergence.** We agree that more experiments in easier or harder settings could illustrate how different rates of convergence affect our method. However, in practice, one would not train a model from scratch on the labeled data, but use the labeled data to fine-tune existing pretrained models. In fact, our framework is able to learn a labeling policy or predictor using outputs of the pretrained machine learning model we are calibrating, making the learning task much easier. The regret bound elucidates how the error of the label policy and predictor propagates into the growth rate. We do agree that the efficacy of these pretrained models would be different for different tasks, and we will be sure to include empirical analysis of how the accuracy pretrained (or learned) label policies and predictors affect the accuracy of our $\hat{\beta}$ estimate.
>
> Our responses to your questions are as follows.
>
> 1. **Does the convergence rate of the estimation terms largely dominate the $O(\log(T))$ rate from the $G^\beta$-contribution?**
> In terms of asymptotic rates, we agree with your point that the $O(\log(T))$ regret incurred by online Newton step (or other online learning algorithms) will be dominated by a typical estimation rate that scales with $O(1 / \sqrt{T})$ if we are training the label policy and predictor completely from scratch. When our pretrained machine learning models are already accurate, however, we can have fast rates. An example of this is if we assume that the optimal policy/predictor lies in a class of functions that is formed by a linear combination of multiple existing pretrained models we have access to (or we are only interested in comparing against the best linear combination). In that case, the estimation error is the regret of an online linear regression algorithm, and it will have $O(\log(T) / T)$ convergence (although we would be directly using the $G^\beta$ regret result, rather than the estimation error result). So our convergence rate does depend on the assumption we make about the accuracy of our pretrained models.
> 2. **Obtaining an adaptive rate.** Our result is already instance-adaptive in the sense that the optimal $G^\beta$ is adaptive to the $X$-conditional variance of the risk, i.e., it is lower bounded by a term that increases as $\mathbb{E}[\sigma_\beta(X)]$ decreases.
> It is possible that when $\mathbb{E}[\sigma_\beta(X)]$ is low, the optimal predictor is estimated more quickly, and we believe exploration of adaptive convergence rates of different estimators would be an interesting problem to solve in future work. Our focus in this work is to show that the $G^\beta$ regret depends on estimator error of the policy and predictor in a direct way, and hence having pretrained models that can approximate those well will also improve the efficiency of estimating $\beta^*$.

---

> > ### Comment · Reviewer_1rh6 · 2024-08-12
> >
> > I acknowledge the authors' response, and thank them for their detailed answers to my points and questions.
> >
> > While it appears that we've all had some difficulty related to the presentation, I feel that my understanding of the method and its utility/scope has been greatly enhanced by this discussion, and am intrigued to see this line of work built upon from both practice and theory (it's a bias, but the adaptive rates question certainly appeals). Furthermore, I do see this work as a very nice potential bridge between theory and practice, and would like to see more work in machine learning with this kind of scope.
> >
> > I would advise a more targeted rework of the presentation of results (for example, by incorporating the concrete suggestions by XpAy) such that researchers with a more practical leaning can get what they need out of it. Otherwise, I stand by my assessment of the paper, and recommend it for acceptance.

---

### Author Rebuttal · Authors · 2024-08-07

We have made point-by-point responses to each review, and posted our rebuttals to each review below. We would also like to note the following.

1. We appreciate the suggestions and concerns the reviewers have brought up about the paper. We will incorporate their suggestions for clarification and make our introduction and setup more accessible to the reader.
2. We will also provide more concrete application examples in the text. We address this in our response to reviewer qKw8, which we replicate here.
    - *Reduce query cost in medical imaging.* A medical imaging system that outputs scores for each pixel of image that determines whether there is a lesion or not would want to utilize labels given by medical experts for unlabeled images from new patients. Since the cost of asking experts to label these images is quite high, one would want to query experts efficiently, and only on data that would be most helpful for reducing the number of highlighted pixels.
    - *Domain adaptation for behavior prediction.* One reason we would want online calibration in a production setting is that we may have much different distribution of data that we do not have access to before deployment. For example, during a navigation task for a robot, we may want to predict the actions of other agents and avoid colliding into them when travelling between two points [1]. Since agents may behave differently in every environment, it makes sense to collect the behavior data in the test environment and update the behavior prediction in an online fashion to get accurate predictions calibrated for specifically the test environment.
    - *Safe outputs for large language models (LLMs).* One of the goals with large language models is to ensure their responses are not harmful in some fashion (e.g., factually wrong, toxic, etc.). One can view this as outputting a prediction set for the binary label set of $Y \in \\{\texttt{harmful},\ \texttt{not harmful}\\}$. Many pipelines for modern LLMs include some form of a safety classifier, which scores the risk level of an output, and determines whether it should be output to the user or not [2, 3], or a default backup response should be used instead. One would want to label production data acquired from user interaction with the LLM and used to calibrate cutoff for the scores that are considered low enough for the response to be allowed through.
3. We appreciate the reviewers' concern about experiments and we will aim to provide more experiments in the final version (i.e., QA factuality task for LLMs, and image labeling for COCO-MS dataset).

4. **Comparison to active learning.** We briefly summarize the differences and similarities between active learning and our problem. Our problem objective and the methods use to prove their validity are different from typical active learning methods. Active learning (stream-based and pool-based) aims to minimize label queries and still learn the best possible machine learning predictor. Guarantees in this area usually are model-based or learning theoretic, i.e., they propose a model update/selection procedure and query procedure that minimizes the true risk of the model over a known or arbitrary function class, and derive results using a notion of class complexity (when one is developing a procedure that agnostic to the exact function class), or the methods are evaluated empirically for specific models, without guarantees. In contrast, our procedure tunes a calibration parameter that can be wrapped around any black-box model to provide a statistically rigorous risk guarantee. As a result, it means that other types of querying strategies that are deterministic (e.g., disagreement based, diversity based, etc.) cannot be directly imported to our problem setting, since the statistical guarantees we derive require that our queries are probabilistic. Further, we do not think existing active learning necessarily tackle the same objective, since they focus primarily on optimizing the performance of a classifier, rather than guaranteeing risk control while calibrating a parameter. Further development of how to leverage active learning methods in our setting is a fruitful direction for future work.

**References**

[1] J. Lekeufack, A. N. Angelopoulos, A. Bajcsy, M. I. Jordan, and J. Malik. Conformal Decision Theory: Safe Autonomous Decisions from Imperfect Predictions. arXiv:2310.05921, 2024.

[2] T. Markov, C. Zhang, S. Agarwal, F. E. Nekoul, T. Lee, S. Adler, A. Jiang, and L. Weng. A Holistic Approach to Undesired Content Detection in the Real World. *AAAI*, 2023

[3] L. Hanu and Unitary team. Detoxify. Github. https://github.com/unitaryai/detoxify, 2020.

---

### Author Response · Authors · 2024-08-14
**Rebuttal recap**

We thank the reviewers for reading our rebuttals and recommending or approving acceptance of our paper. We also appreciate the thoughtful feedback that we have received throughout the review process, and we will be sure to incorporate this in the ways we've described in our rebuttal.

---

### Decision · Program_Chairs · 2024-09-25

**Decision:**

Accept (poster)

**Comment:**

The paper focuses on extending the concept of Risk Controlling Prediction Sets (RCPS) to a sequential setting, where data is collected over time. This extension ensures that risk guarantees are anytime-valid, i.e. simultaneously hold at all time steps. The authors also introduce a framework for active labeling, allowing a model to selectively query labels within a predefined budget while maintaining low risk. They further propose methods to improve the utility of RCPS by leveraging model predictions to estimate conditional risks. The paper characterizes optimal labeling policies under a fixed budget, presents a regret analysis linking estimation errors to the RCPS framework, and demonstrates the effectiveness of these methods through empirical results, showing that their approach requires fewer labels to achieve higher utility compared to traditional labeling strategies.

The reviewers were generally positive in their evaluation of the paper. I recommend accepting it, provided that the authors incorporate the reviewers' suggestions, particularly those related to improving the presentation, in the final version.